# LLM-I: LLMs are Naturally Interleaved Multimodal Creators

## Abstract

We propose LLM-Interleaved (**LLM-I**), a flexible and dynamic framework that reframes interleaved image-text generation as a tool-use problem. LLM-I is designed to overcome the "one-tool" bottleneck of current unified models, which are limited to synthetic imagery and struggle with tasks requiring factual grounding or programmatic precision. Our framework empowers a central LLM or MLLM agent to intelligently orchestrate a diverse toolkit of specialized visual tools, including online image search, diffusion-based generation, code execution, and image editing. The agent is trained to select and apply these tools proficiently via a Reinforcement Learning (RL) framework that features a hybrid reward system combining rule-based logic with judgments from LLM and MLLM evaluators. Trained on a diverse new dataset using four different model backbones, LLM-I demonstrates state-of-the-art performance, outperforming existing methods by a large margin across four benchmarks. We also introduce a novel test-time scaling strategy that provides further performance gains.

## 1 Introduction

AI is shifting from single-modality systems to multimodal ones that can process mixed data like text, images, and sound. A key frontier is interleaved image-text generation (Ge et al., 2024; Tian et al., 2024; Xie et al., 2025; Zhou et al., 2025b; Xia et al., 2025; Chen et al., 2025): producing a coherent, alternating sequence of text and images from a single prompt. However, the task is technically demanding, requiring high-fidelity text and images with strict cross-modal consistency. This involves maintaining narrative coherence, consistent visual style and entities, and strong semantic alignment between each image and its accompanying text.

To address these challenges, the research community has largely converged on two dominant architectural paradigms for interleaved image-text generation. The first, a two-stage or compositional approach, leverages the distinct strengths of separate, state-of-the-art models (Zhou et al., 2025b) or add decoders (Ge et al., 2024) after the text generation. In this paradigm, a powerful LLM, such as GPT-4o (Hurst et al., 2024), acts as a high-level reasoning engine. It interprets the user's request to produce a sequence of textual narratives, which are then passed to a separate, high-fidelity text-to-image diffusion model, such as DALL-E (Betker et al., 2023) or Seedream (Gao et al., 2025), for visual synthesis. However, it often suffers from a "semantic gap", where the LLM's textual representation of a desired image may not perfectly align with the diffusion model's interpretation, leading to inconsistencies. Furthermore, these systems lack flexibility, as they are typically restricted to generating a fixed number of images per response.

Seeking to close this gap and achieve greater architectural elegance, a significant research effort has been directed towards developing unified, end-to-end models (Xie et al., 2025; Zhou et al., 2025a; Deng et al., 2025) that handle both multimodal understanding and generation within a single, integrated framework. Despite their notable advancements, current unified models for interleaved generation suffer from a critical and largely unaddressed limitation: the "one-tool" bottleneck. While these unified models excel at generating novel, high-fidelity synthetic imagery from textual prompts, they are inherently ill-suited for tasks that require factual grounding such as real-world images or programmatic precision such as data analysis and visualizations. This architectural commitment creates a rigid system that forces a single tool to solve all visual generation problems, regardless of its suitability. This "one-tool" bottleneck reflects a deeper paradigm choice in AI development:

the pursuit of an "omniscient solver" that embeds all knowledge within its parameters, rather than a "proficient tool-user" that knows how to leverage external resources. The latter approach is inherently more flexible, scalable, and robust. A tool-augmented system can be easily updated with new capabilities by simply adding a new tool to its repertoire, whereas a monolithic model requires complete and computationally prohibitive retraining to acquire new skills.

In this paper, we introduce LLM-Interleaved (**LLM-I**), a flexible and dynamic framework that employs an LLM or MLLM as an agentic planner. This central agent leverages its sophisticated reasoning and multimodal understanding capabilities to intelligently orchestrate a diverse suite of external, specialized visual tools for image generations. Our framework equips the central agent with a toolkit of four distinct and complementary visual tools which are *online image search, diffusion-based generation, code generation and execution*, and *image edit tool*. To ensure the agent uses these tools proficiently, we develope a Reinforcement Learning (RL) framework that incorporates a hybrid reward design, combining rule-based rewards and LLM and MLLM judges. We build a diverse dataset for training and evaluate LLM-I using four different backbone models, finding that it outperforms state-of-the-art methods by a large margin across four benchmarks. Additionally, we propose a novel test-time scaling strategy that improves performance even further.

We summarize our key contributions as follows:

1. **Novel Framework for Interleaved Generation**: We propose a new and flexible paradigm, LLM-I, for interleaved image-text generation. Our framework recasts the LLM/MLLM not as an end-to-end generator but as an intelligent agent that orchestrates a toolkit of external, specialized visual models. This approach decouples high-level reasoning from low-level synthesis, enabling unprecedented flexibility and context-appropriateness in the generated multimodal content.

2. **New Dataset and Benchmark**: We introduce a diverse dataset and difficult benchmark for interleaved image-text generation. Our work moves beyond the scope of previous datasets by requiring multiple forms of images, including retrieved real-world photos, synthetic visuals, and programmatic visualizations.

3. **Strong Performance**: LLM-I outperforms previous SOTA methods by a large margin across four benchmarks. Through test-time scaling, the performance is further improved.

## 2 METHODOLOGY

### 2.1 TOOL USAGE

#### 2.1.1 MOTIVATION

As we discussed above, current methods (Chern et al., 2024; Wu et al., 2024; Zhou et al., 2025a; Xie et al., 2025) are locked into a single mode of creation, limiting the scope, factuality, and utility of the narratives they can produce. It is instructive to consider how humans approach a similar task, such as authoring a blog post or a technical report. When a writer needs to insert an image, they rarely create it from scratch. Instead, they act as an intelligent agent, selecting the best external tool for the job. If they need a picture of the Eiffel Tower, they use a search engine to find a real photograph. To display quarterly sales data, they would use software like PowerPoint or a coding library to generate a precise chart. They might also use an image editing tool like Photoshop to make adjustments, such as cropping a photo, adjusting its colors, or adding annotations to highlight key information. This human workflow is not monolithic; it is dynamic, flexible, and tool-centric. The writer's primary skill is not drawing but reasoning and orchestrating a diverse set of specialized tools to achieve their goal.

Therefore, we argue that a paradigm that mimics this human-like, tool-using strategy holds significant advantages over current monolithic models. An AI system that can intelligently invoke external tools is inherently more flexible, scalable, and robust. It can ground its generations in factual reality by searching the web, provide precise data visualizations through code execution, and still retain the ability for other tasks. This approach directly overcomes the "one-tool" bottleneck, moving beyond the limited "omniscient solver" paradigm towards a more powerful and practical "proficient tool-user".

### 2.1.2 TOOLKIT

Motivated by this insight, we introduce a flexible and dynamic framework where an LLM or MLLM serves as an agentic planner. We empower this central agent to intelligently orchestrate a suite of distinct visual tools to construct rich, interleaved content. Specifically, our framework equips the agent with capabilities for online image search, diffusion-based generation, code execution for data visualization, and image editing.

1. **Online Image Search**: Invoked for requests demanding factual grounding, such as specific real-world entities, landmarks, or current events. This tool ensures visual authenticity and provides access to up-to-date information beyond the model's training data cutoff. In our paper, we use Google Search API (Google, 2025b).

2. **Diffusion-based Generation**: Selected for tasks requiring the creative synthesis of novel or abstract concepts, or complex compositions that do not exist in reality. We support Seedream 3.0 (Gao et al., 2025) in our paper.

3. **Code Execution**: Utilized primarily for generating data visualizations like charts, graphs, and plots from structured data. We use Python as the programming language and build a controlled sandbox environment.

4. **Image Editing**: Engaged to perform modifications on existing visual content, whether inputted, retrieved or generated. We support Seededit 3.0 (Wang et al., 2025) in our project.

### 2.1.3 HOW TO CALL A TOOL?

To empower the LLM to dynamically orchestrate our suite of visual tools, we design a robust and flexible tool invocation framework. Instead of complex, multi-turn interactions or fine-tuning on specific API call formats, our approach is guided by a system prompt that instructs the model to embed a specific placeholder tag wherever visual element is required in the narrative. This method allows the LLM to autonomously decide when and how to use a tool within a single generative pass.

The core of our framework is the structured tag, `<imgen>{...}</imgen>`, which encapsulates all the necessary information for generating or retrieving an image. When the LLM determines that an image is needed, it generates this tag in the following JSON-like format:

```
<imgen>{"source":"<source type>", "description":"<general title>",
"params":{...}}</imgen>
```

For search, the params contains a single key *query* which holds a practical and concise search string for a web image search engine. For diffusion, it contains the key *prompt*, which provides a descriptive text prompt for the generative model. For code, it contains the key *code* which holds the raw Python code snippet required to generate a plot or visualization. For edit, it contains two keys, *img index*, the 0-based index of a previously image in the sequence to be modified, and *prompt*, a textual instruction describing the desired edit.

When the tag is detected in the generated sequence, a parser processes this output, identifies each tag, and dispatches a call to the corresponding external tool using the provided parameters. The tag is then replaced in the text with the image returned by the tool, resulting in the final, seamless multimodal document.

## 2.2 RL RECIPE

### 2.2.1 DATASET CONSTRUCTION

To train our model, we construct a high-quality RL dataset of approximately 4,000 samples with a "tool-oriented" design philosophy. The dataset is bifurcated into text-only and text-and-image inputs. The generation process is automated to produce implicit prompts that describe a desired outcome without naming the specific tool required, thereby encouraging the model to reason about tool selection.

Each sample undergoes a rigorous validation pipeline to ensure high quality and fidelity. A critical feature of this dataset is the annotation of each prompt with an image number constraint, which guides the RL training process by specifying the rules for image generation. This constraint falls

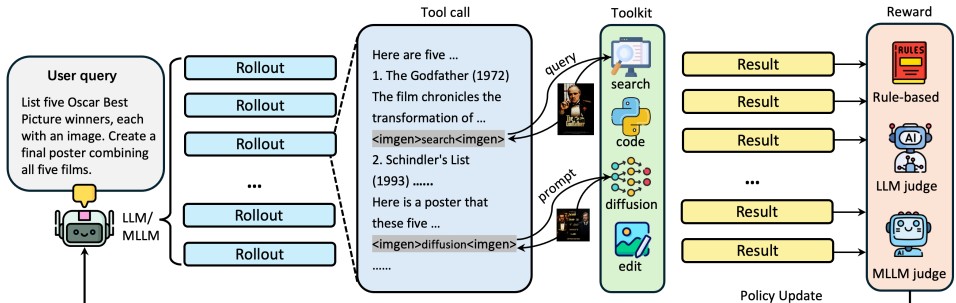

Figure 1: Overview of the LLM-I framework.

into one of four categories: images are disallowed (-1), their use is unconstrained (0), a precise quantity n is required ($n > 0$), or at least one image is mandatory (`Inf`). Further details on the generation and validation process are provided in the Appendix B.

### 2.2.2 REWARD

With the instruction dataset in place, we employ an RL strategy to fine-tune the model's ability to appropriately call and parameterize the visual tools. Our approach is distinguished by a multi-faceted reward function that combines deterministic rules $R_{rule}$ with sophisticated judgments from both LLM $R_{llm}$ and MLLM $R_{mllm}$. This composite reward signal not only provides a holistic assessment of the generated output but also decreases reward hacking.

The first component is a deterministic, rule-based reward $R_{rule}$ that enforces adherence to generation constraints and ensures the correctness of the `<imgen>` tag format. In Section 2.2.1, we set a required image number $N_{req}$ for each single item. For categorical constraints, the reward is binary. When images are disallowed ($N_{req} = -1$) or when at least one is required ($N_{req} = \inf$), the model receives a score of 1 for compliance and 0 for violation. When there is no constraint ($N_{req} = 0$), the score is always 1, as any output is considered valid. For quantitative constraints where a precise number of images $n$ is required ($N_{req} = n$), the reward is designed to penalize both under- and over-generation:

$$R_{rule} = \begin{cases} \frac{N_{gen}}{N_{req}} & \text{if } 0 \leq N_{gen} \leq N_{req} \\ \max(0, 1 - \alpha \cdot (N_{gen} - N_{req})) & \text{if } N_{gen} > N_{req} \end{cases} \quad (1)$$

where $N_{gen}$ is the number of generated images in the model response and $\alpha$ is the penalty factor of extra images which is set to 0.3 by default.

The second component $R_{llm}$ leverages an external LLM as a judge to assess the quality of the language and the logic of the tool invocation. This judge evaluates two criteria on a 1-to-5 scale: (i) the fluency, coherence, and relevance of the textual narrative, and (ii) the quality of the tool-use tags, including the naturalness of their placement and the semantic appropriateness of the chosen source and params.

The third reward component $R_{mllm}$ employs an MLLM to evaluate the final interleaved output. After the images are generated and integrated, this judge scores three key aspects of multimodal quality on a 1-to-5 scale: (i) the technical and aesthetic quality of the image itself, (ii) the semantic alignment between the image and its surrounding text, and (iii) the relevance of the image to the overall task objective.

The scores from the LLM and MLLM judges are normalized to a [0, 1] range. The final reward signal $R$, is then composed from all three components. Notably, the rule-based reward $R_{rule}$, acts as a multiplicative gate on the MLLM reward $R_{mllm}$. This formulation ensures that visual quality is considered only if the model has first satisfied the explicit image count constraint. The composite reward is thus defined as:

$$R = w_{rule}R_{rule} + w_{llm}R_{llm} + w_{mllm}R_{mllm}R_{rule} \quad (2)$$

where $w_{rule}$, $w_{llm}$, and $w_{mllm}$ are the trade-offs between the three losses.

Figure 2: Overview of test-time scaling framework for LLM-I.

## 2.3 TEST-TIME SCALING

To further enhance the performance of our agentic framework, we introduce a *test-time scaling* (Snell et al., 2025; Muennighoff et al., 2025) paradigm that leverages additional computational resources during inference. The goal is to improve both the reliability of tool usage and the overall quality of the final multimodal response. The workflow is illustrated in Figure 2.

Given a user query, the model first generates multiple complete candidate responses through stochastic sampling. Each candidate may contain tool calls (*e.g.,* search, diffusion, code, or editing), interleaved with natural language. The initial $n$ candidates are passed through a "Tool Call Check" filter. This stage validates the structural integrity and executability of the tool invocations. Responses with malformed or failed tool calls are discarded. From the pool of successful candidates, a selector model (an LLM/MLLM) evaluates their overall quality and relevance to the prompt, selecting the top-k most promising responses for further enhancement. Then, each of the $k$ selected candidates undergoes a targeted enhancement process based on the type of tool used. When a response requests an image, we concurrently query both the online image search module and the diffusion model. The resulting candidates are evaluated by an MLLM, which selects the most semantically aligned option. If code execution fails, the erroneous code and the associated error message are provided to a model. The model revises the code, which is then re-executed in a sandboxed environment until a valid visualization is obtained or exceeding a fixed number of attempts. After the enhancement, the $k$ refined interleaved multimodal responses are passed to an MLLM for polishing. This step improves the coherence and alignment between modalities, ensuring that visual outputs are seamlessly integrated with textual explanations. Finally, a selector model ranks the polished candidates and chooses the single best response as the final output.

## 3 BENCHMARK

To rigorously evaluate models on generating sophisticated, interleaved text-image reports, we introduce a new benchmark LLMI-Bench. It is designed to address two primary limitations of existing benchmarks (Liu et al., 2024; Zhou et al., 2025b; Chen et al., 2025): (1) overly simplistic prompts that require only decorative, low-information images, and (2) unreliable, subjective evaluation protocols that rely on forgiving LLM judges.

Our benchmark overcomes these challenges through two innovations. First, we reframe the generation task as a "mini-project," where prompts provide specific data or context that necessitates the creation of images with high informational value (*e.g.,* data visualizations, scientific illustrations). In this paradigm, images are an indispensable, synergistic component of the report. Second, we replace broad rubrics with a sample-specific, objective evaluation protocol. For each task, we design a unique set of concrete, verifiable criteria. An LLM evaluator then scores the output against these specific rules, transforming the assessment from a subjective judgment into a more objective and reliable measurement. Further details can be found in Appendix C.

## 4 EXPERIMENTS

### 4.1 SETUP

**Data and Benchmarks:** We train our model using the data constructed in Section 2.2.1, which is split into a training set and an in-domain test set containing over 200 samples. For a comprehensive

Table 1: Results on the OpenING benchmark. † is evaluated with text-only input samples. IT Coherency refers to image-text conherency and MS consistency means multi-step consistency. Qwen series are all evaluated with tools.

| Model | Completeness | Quality | Richness | Correctness | Human Alignment | IT Coherency | MS Consistency | Overall |
|---|---|---|---|---|---|---|---|---|
| GPT4o+DALLE3 | 8.66 | 8.01 | 7.42 | 7.98 | 8.77 | 8.15 | 8.38 | 8.20 |
| Gemini+FLUX | 7.58 | 7.26 | 6.48 | 7.03 | 7.98 | 6.98 | 7.33 | 7.23 |
| NExT-GPT | 3.89 | 4.25 | 3.35 | 3.61 | 5.35 | 3.32 | 3.85 | 3.95 |
| Show-o | 4.37 | 4.79 | 3.83 | 3.76 | 5.78 | 4.04 | 4.33 | 4.41 |
| SEED-X | 5.65 | 6.07 | 4.92 | 5.77 | 7.03 | 5.72 | 5.72 | 5.84 |
| Anole | 6.27 | 6.02 | 5.28 | 5.06 | 6.91 | 4.90 | 5.81 | 5.75 |
| Qwen2.5-VL-7B | 2.97 | 3.90 | 2.50 | 3.07 | 4.37 | 2.03 | 3.82 | 3.24 |
| Qwen2.5-VL-32B | 6.78 | 6.82 | 5.89 | 6.34 | 7.25 | 5.69 | 7.15 | 6.56 |
| MLLM-I-7B | 6.00 | 6.75 | 5.53 | 5.85 | 7.24 | 5.85 | 6.50 | 6.25 |
| MLLM-I-32B | 8.35 | 8.07 | 7.48 | 7.79 | 8.44 | 7.35 | 8.38 | 7.98 |
| Qwen3-4B-Instruct† | 6.26 | 6.88 | 5.55 | 6.09 | 6.95 | 5.11 | 6.86 | 6.24 |
| Qwen3-30B-Instruct† | 8.05 | 7.63 | 7.09 | 7.56 | 8.12 | 6.90 | 8.13 | 7.64 |
| LLM-I-4B† | 8.63 | 8.03 | 7.54 | 8.03 | 8.69 | 7.87 | 8.45 | 8.18 |
| LLM-I-30B† | 9.19 | 8.44 | 8.08 | 8.61 | 8.99 | 8.40 | 8.91 | 8.66 |

evaluation, we utilize this in-domain test set along with three out-of-domain (OOD) benchmarks. On the in-domain set, we employ the same metrics used during training: a rule-based metric, LLM-based judgments, and MLLM-based judgments. For OOD evaluation, we use the public OpenING benchmark (Zhou et al., 2025b), which has 5,400 samples (2,491 text-only and 2,909 multimodal inputs), and adopt the seven metrics from the original paper. Besides, we use the public benchmark ISG (Chen et al., 2025), which has over 1,000 samples, and adopt the four metrics from the original paper. Additionally, we introduce our novel and much more difficult LLMI-Bench, whose manageable size enables a multifaceted evaluation through a rubric-based scoring rate from GPT-4o, a rule-based metric to measure tool-call success, and a rigorous human evaluation. For this human assessment, we design a five-point Likert scale with detailed criteria for each point and calculate the final metric as the average overall scoring rate.

Table 2: Results on the LLMI-Bench. † is evaluated with text-only input samples. Tool Acc refers the success rate of tool invocation.

| Model | Rubric | Human | Tool Acc | Overall |
|---|---|---|---|---|
| GPT-5 wTool | 53.8 | 48.3 | 28.1 | 43.4 |
| GPT-4o wTool | 70.4 | 62.8 | 67.9 | 67.0 |
| Anole | 27.4 | 18.2 | - | 22.8 |
| Qwen2.5-VL-7B wTool | 28.5 | 19.3 | 44.3 | 30.7 |
| Qwen2.5-VL-32B wTool | 58.9 | 51.1 | 93.4 | 67.8 |
| Qwen2.5-VL-72B wTool | 73.1 | 59.8 | 60.1 | 64.3 |
| MLLM-I-7B | 67.1 | 61.9 | 97.4 | 75.5 |
| MLLM-I-32B | 92.5 | 82.1 | 99.2 | 91.3 |
| Qwen3-4B-Instruct wTool† | 73.6 | 62.3 | 68.7 | 68.2 |
| Qwen3-30B-Instruct wTool† | 81.4 | 69.2 | 83.1 | 77.9 |
| LLM-I-4B† | 88.9 | 72.9 | 100.0 | 82.3 |
| LLM-I-30B† | 94.8 | 83.3 | 100.0 | 92.7 |

Table 3: Results on the ISG benchmark. † is evaluated with text-only input samples.

| Model | Structural | Holistic | Block | Image |
|---|---|---|---|---|
| Show-o | 0.295 | 2.329 | 1.962 | 0.078 |
| Anole | 0.000 | 2.810 | - | - |
| Gemini+SD3 | 0.385 | 5.827 | 3.081 | 0.113 |
| ISG | 0.871 | 6.262 | 5.515 | 0.574 |
| Qwen2.5-VL-7B | 0.085 | 4.932 | 1.152 | 0.016 |
| Qwen2.5-VL-32B | 0.221 | 6.354 | 2.105 | 0.088 |
| MLLM-I-7B | 0.607 | 6.381 | 3.584 | 0.274 |
| MLLM-I-32B | 0.776 | 8.112 | 5.722 | 0.419 |
| Qwen3-4B† | 0.068 | 5.621 | 1.621 | 0.086 |
| Qwen3-30B† | 0.267 | 7.848 | 3.811 | 0.267 |
| LLM-I-4B† | 0.881 | 8.413 | 7.701 | 0.511 |
| LM-I-30B† | 0.971 | 8.492 | 8.291 | 0.618 |

**Training:** We conduct experiments using four different backbones, covering both LLMs and MLLMs. They include Qwen3-4B-Instruct, Qwen3-30B-Instruct (MoE model), Qwen2.5-VL-7B, and Qwen2.5-VL-32B. For MoE model, we use GSPO (Zheng et al., 2025) as the RL algorithm while for others we use GRPO (Shao et al., 2024). We use a batch size of 32 with a cosine learning rate scheduler where the initial learning is set to 1e-6, minimum learning rate ratio is set to 0.01, and the warm-up step is 5. Following Yu et al. (2025), we use the token-level loss. For GSPO, the clipping ratios are set to 3e-4 (low) and 4e-4 (high). For judgement, we use Qwen3-235B-Instruct-2507 (Yang et al., 2025) as the LLM judge and Qwen2.5-VL-72B-Instruct (Bai et al., 2025) as the MLLM judge. The reward trade-off coefficients are set to $w_{rule} = 0.2$, $w_{llm} = 0.5$, and $w_{mllm} = 0.3$.

## 4.2 MAIN RESULTS

Tables 1, 2, 3 and 4 present the detailed results of our model across four distinct benchmarks. Our evaluation compares LLM-I against a diverse set of baselines categorized into three main types: (i) two-stage or compositional methods such as GPT-4o+DALLE-3, Gemini+FLUX/Stable Diffusion 3 (SD3) (Esser et al., 2024), NExT-GPT (Wu et al., 2024), SEED-X (Ge et al., 2024), and ISG (Chen

Table 4: Results on the test set of the dataset. † is evaluated with text-only input samples. IT means image-text and IQ means image-question. Qwen series are all evaluated with tools.

| Model | Image num | Text Quality | Tag Quality | Image Quality | IT Alignment | IQ Alignment | Overall |
|---|---|---|---|---|---|---|---|
| Qwen2.5-VL-7B | 13.2 | 3.1 | 1.5 | 3.7 | 2.8 | 2.9 | 25.9 |
| Qwen2.5-VL-32B | 54.3 | 3.8 | 2.0 | 3.8 | 3.6 | 3.5 | 52.7 |
| MLLM-I-7B | 88.7 | 4.0 | 3.4 | 3.9 | 3.9 | 3.7 | 70.6 |
| MLLM-I-32B | 95.1 | 4.7 | 4.3 | 4.1 | 4.2 | 4.3 | 85.2 |
| Qwen3-4B-Instruct† | 46.5 | 4.5 | 2.9 | 4.0 | 3.9 | 3.9 | 57.7 |
| Qwen3-30B-Instruct† | 55.3 | 4.8 | 4.0 | 3.9 | 3.9 | 3.9 | 68.7 |
| LLM-I-4B† | 88.6 | 4.8 | 4.6 | 4.2 | 4.2 | 4.3 | 85.2 |
| LLM-I-30B† | 93.0 | 4.9 | 4.8 | 4.3 | 4.6 | 4.6 | 89.9 |

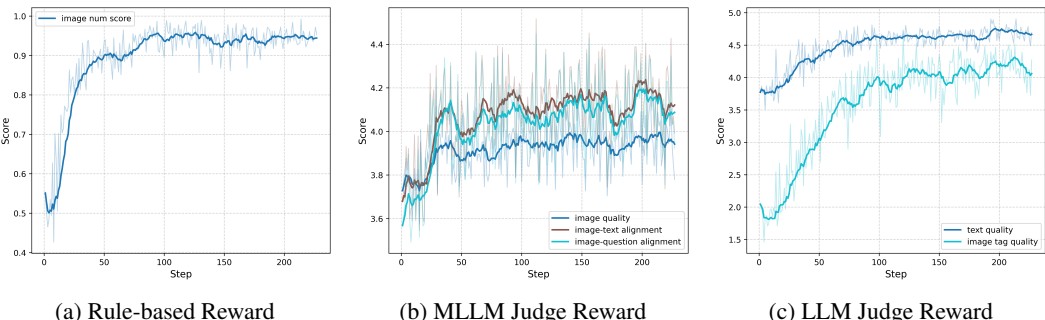

(a) Rule-based Reward  (b) MLLM Judge Reward  (c) LLM Judge Reward

Figure 3: Reward curve during RL training of Qwen2.5-VL-32B.

et al., 2025); (ii) unified models including Show-o (Autoregressive+Diffusion) (Xie et al., 2025) and Anole (Pure Autoregressive) (Chern et al., 2024); and (iii) tool-augmented methods, featuring GPT-5 (OpenAI, 2025b) and GPT-4o with a suite of tools that includes search, diffusion, code, and editing capabilities. Across all four benchmarks, we observe that LLM-I exhibits SOTA performance, consistently and significantly outperforming baseline models.

In the general qualitative assessment shown in Table 1 and 3, the entire LLM-I family shows highly competitive performance, outperforming other leading models and unified approaches, which underscores our model's robustness in generating complete, high-quality, and well-aligned content. This superiority is even more pronounced on specialized benchmarks. On the LLMI-Bench evaluation in Table 2, LLM-I models drastically outperform dedicated tool-using agents, including GPT-4o w/Tool and the anticipated GPT-5 w/Tool. This success is largely attributable to our model's exceptional tool invocation capabilities, with LLM-I-4B and LLM-I-30B achieving a perfect 100.0 Tool Accuracy. Additionally, we present the metrics dur-

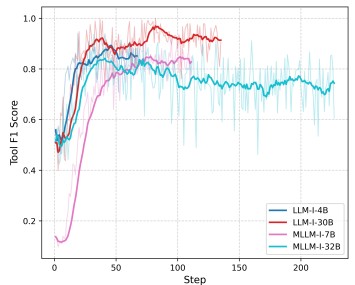

Figure 6: Tool F1 score curve during RL training.

ing the RL training in Figure 3. We can observe that with the RL training, the instruction following ability, writing ability and the ability to find images of the model all increases, indicating the effectiveness of RL training.

Furthermore, we present two examples in Figure 4 and 5. From the examples, we can observe that LLM-I can intelligently invoke different kinds of tools for image presentation. This shows great advantages over previous methods when requiring real and precise images. In the dataset construction stage in Section 2.2.1, we define a target tool list for each data item which is verfied by Gemini2.5 Pro and GPT-4o. To further validate the "intelligence" of tool invocation, we visualize the tool F1 score in Figure 6 which evaluates the precision and recall of different tools during the training process. From the figure, the F1 score steadily improves during the RL process, indicating that the model becomes increasingly adept at selecting appropriate tools according to the given context. Notably, no explicit reward is assigned to tool usage; the improvement arises naturally during RL training. This finding suggests that RL not only encourages tool invocation but also enhances the model's ability to make smarter tool choices for achieving better image–text alignment.

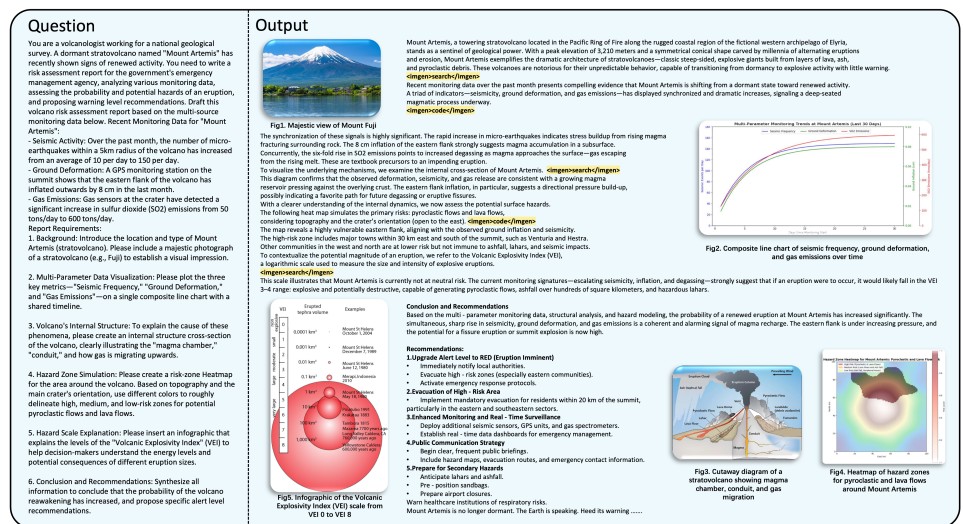

Figure 4: Example generated by LLM-I on LLMI-Bench. Some text is omitted due to space constraints.

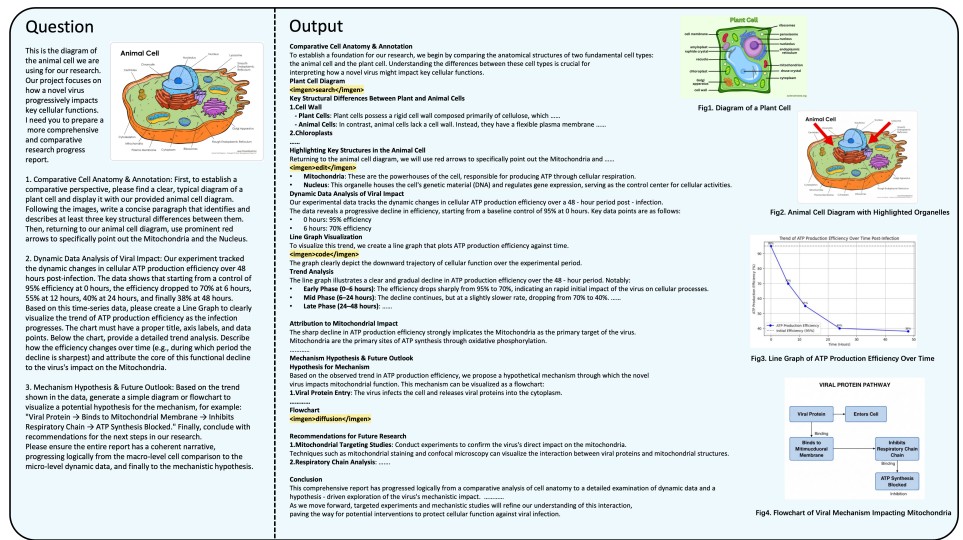

Figure 5: Example generated by MLLM-I on LLMI-Bench. Some text is omitted due to space constraints.

## 4.3 TEST-TIME SCALING

Table 5 presents the performance of our proposed test-time scaling strategy. As detailed in Section 2.3, this strategy comprises four stages: initial top-k selection, tool enhancement, polishing, and final selection. In our experiments, we set the initial selection parameter $k$ as 4 and sample 2 images for both the search and diffusion tools. The Qwen2.5-VL-72B model serves as both the selector and the polisher.

The results in Table 5 demonstrate the efficacy of our approach. The initial top-k selection and tool enhancement stages substantially boost performance. The subsequent polishing stage also provides improvement. By integrating all four stages, our model surpasses the performance of its 30B counterpart, validating the effectiveness of our test-time scaling strategy.

Table 5: Results of test-time scaling on LLMI-Bench.

| Model | Rubric | $\Delta$Time |
|---|---|---|
| LLMI-4B | 88.9 | 0 |
| - w stage1 | 91.2 | <1s |
| - w stage2 | 91.4 | ~1s |
| - w stage3 | 89.4 | ~16s/it |
| w full TTS | 95.1 | ~20s/it |
| LLM-I-30B | 94.8 | 0 |

Table 6: Results of reward ablation experiments on the OpenING benchmark.

| Model | Completeness | Quality | Richness | Correctness | Human Align. | IT Conhe. | MS Consis. | Overall |
|---|---|---|---|---|---|---|---|---|
| Qwen3-4B-Instruct | 6.26 | 6.88 | 5.55 | 6.09 | 6.95 | 5.11 | 6.86 | 6.24 |
| LLM-I-4B | 8.63 | 8.03 | 7.54 | 8.03 | 8.69 | 7.87 | 8.45 | 8.18 |
| w/o rule-based reward | 4.22 | 6.55 | 3.93 | 4.55 | 5.68 | 2.34 | 6.05 | 4.76 |
| w/o LLM judge | 8.29 | 7.77 | 7.23 | 7.69 | 8.38 | 7.44 | 8.18 | 7.85 |
| w/o MLLM judge | 8.17 | 7.66 | 7.20 | 7.60 | 8.23 | 7.39 | 8.04 | 7.76 |

We also analyze the computational overhead introduced by this strategy. A key advantage is that tool invocations can be processed in parallel. Consequently, the primary overhead consists of only four additional forward passes from the selector/polisher model. The selection process is particularly efficient, as the model only needs to output the optimal index rather than generating a full response. In contrast, the polishing stage is the most time-consuming, as it requires rewriting the entire response.

## 4.4 ABLATION STUDY

**Effectiveness of the Reward Design.** We conduct an ablation study to evaluate the individual contributions of our three reward components: a rule-based reward, an LLM judge, and an MLLM judge. The results are presented in Table 6. The full LLM-I-4B model, trained on all three rewards, establishes a strong baseline with an overall score of 8.18, demonstrating the effectiveness of the combined reward strategy.

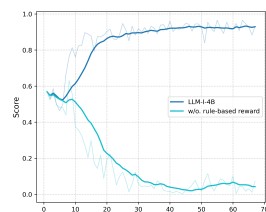

Figure 7: The rule-based reward curve during RL training.

Particularly, the removal of the rule-based reward proves to be the most detrimental, causing the overall score to plummet to 4.76. As shown in Figure 7, without the rule-based reward, the model will not generate the image to obtain a high score. Comparatively, the performance drop is less severe when removing either the LLM or MLLM judge because their evaluation capabilities likely overlap. Both of these judges assess more nuanced, qualitative aspects of the output like text and image quality. Ultimately, the study confirms that while the rule-based reward provides an essential foundation, the synergistic combination of all three rewards is necessary to achieve the model's peak performance.

**Tools.** To assess the contribution of individual tools, we perform a tool ablation study and report the results in Table 7. The results reveal the importance of a comprehensive toolkit, especially for the trained LLMI-4B model. Restricting LLMI-4B to "only diffusion" or "only search" leads to significant performance degradation. This indicates that its high performance is contingent on its ability to flexibly leverage multiple tools.

Table 7: Ablation experiments of the tools on LLMI-Bench.

| Model | Rubric |
|---|---|
| Qwen3-4B-Instruct wTool | 73.6 |
| - only diffusion | 66.5 |
| - only search | 75.2 |
| LLM-I-4B | 88.9 |
| - only diffusion | 76.5 |
| - only search | 77.5 |

Interestingly, the Qwen3-4B model's performance improves to 75.2 when restricted to "only search", surpassing its full-toolkit baseline. This counterintuitive result suggests that while the model benefits from the search tool, it may struggle with tool selection when presented with multiple options when it is not trained to use the tools. Forcing it to use only its most effective tool eliminates potential errors in tool orchestration, thereby improving its overall score.

## 5 CONCLUSION

In this paper, we introduce LLM-Interleaved (LLM-I), a framework that overcomes the "one-tool" bottleneck in interleaved image-text generation by employing an LLM as an agentic planner. This agent dynamically orchestrates a suite of specialized tools, including web search, diffusion models, code execution, and image editing, to create rich multimodal narratives. LLM-I significantly outperforms state-of-the-art methods, demonstrating that powerful LLMs possess a natural, emergent capability for complex multimodal creation when properly augmented. By championing a flexible "proficient tool-user" paradigm, this work paves the way for future research into expanding the agent's toolkit and enhancing its reasoning for more generalist and capable creative AI.

REPRODUCIBILITY STATEMENT

We detail the construction of the dataset, benchmark, and the training information in our paper. Additionally, the code, datasets and benchmark will be open-sourced for reproducibility.

THE USE OF LARGE LANGUAGE MODELS

On top of the training and dataset construction process which we have detailed in the paper, we only use LLMs for paper polishment.

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

## A  RELATED WORK

**Interleaved Image-Text Generation.**  While current MLLMs, such as the QwenVL (Bai et al., 2025) and InternVL (Zhu et al., 2025) series, excel at processing interleaved image-text inputs, they lack the capability for interleaved generation. Two primary approaches have emerged to address this limitation. The first involves leveraging an external image decoder or diffusion model, as seen in models like NExT-GPT (Wu et al., 2024) and SEED-X (Ge et al., 2024). These methods typically optimize a set of learnable visual tokens that serve as input for a diffusion-based image decoder or directly input all the texts into the diffusion model. The second category consists of unified multimodal models that either integrate an autoregressive model with a diffusion model (Zhou et al., 2025a; Xie et al., 2025) or are entirely autoregressive (Team, 2024; Chern et al., 2024) to achieve unified training and alignment. However, a significant drawback of both paradigms is their inherent unsuitability for tasks requiring factual grounding, such as generating photorealistic images of specific entities, or programmatic precision, such as data analysis and visualization. Diverging from these methods, our approach reframes the LLM or MLLM as an agentic planner that orchestrates four external tools. This tool-augmented framework allows for the creation of a wide range of visual content, from photorealistic and creative imagery to accurate data visualizations, thereby overcoming the key weaknesses of prior generative systems.

**Reinforcement Learning.**  RL has become a crucial component in developing the latest generation of large models (Guo et al., 2025a), often yielding superior generalization capabilities compared to purely supervised methods. While Proximal Policy Optimization (PPO) (Schulman et al., 2017) is the most common algorithm for fine-tuning LLMs, its reliance on a value model has spurred the popularity of value-free alternatives like GRPO (Shao et al., 2024) and DAPO (Yu et al., 2025). Although many recent works have successfully applied these algorithms to enhance the reasoning abilities of LLMs and MLLMs (Zheng et al., 2025; Guo et al., 2025b; Hong et al., 2025), our research explores a different direction. Instead of focusing on reasoning, we investigate how RL can be used to improve multimodal alignment, the ability to intelligently use tools, and the overall quality of generated reports.

**Tool Usage of LLMs.**  The ability of LLMs to utilize external tools (Feng et al., 2025; Wu et al., 2025) has significantly expanded their capabilities, transforming them from simple text generators into sophisticated agents capable of reasoning, decision-making, and task automation across various domains. For instance, proprietary models like the OpenAI o3 (OpenAI, 2025c) and DeepResearch (OpenAI, 2025a) model can leverage various tools for web search, code execution, and image processing. Similarly, Gemini 2.5 Pro (Comanici et al., 2025) and its DeepResearch (Google, 2024) can call external tools for functions like code execution, web search, or file processing. In the open-source community, projects such as Search-o1 (Li et al., 2025) and Openthinkimg (Su et al., 2025) have also demonstrated the impressive performance improvements of tool-augmented LLMs and MLLMs. Building on these advancements, RL training can further enhance this capability, enabling an LLM to intelligently select the appropriate tool to use, making it possible to address a wider and more complex range of problems.

## B  DATASET DETAILS

To effectively train our model to master the agentic tool-use framework, we first construct a high-quality RL dataset. The central design philosophy is "tool-oriented", aimed at teaching the model to invoke a diverse set of tools under various constraints. The dataset is bifurcated into two primary categories: text-only inputs and text-and-image inputs.

The generation process is automated using Gemini 2.5 Pro (Comanici et al., 2025). We guide prompt creation through a categorical scaffolding system that defines the target tool(s), a pre-designed specific theme for the tool, an image count which implicitly specifies how many images should be given in the response, and a difficulty level (low, medium, high). A crucial principle is that all generated prompts are implicit; they describe a desired outcome and image number that necessitates a specific tool without ever naming it, thereby encouraging the model to reason about tool selection and image number. To counteract the agent's potential aversion to more error-prone tools during RL (a form of reward hacking), we deliberately increase the representation of prompts requiring code and search, which have higher failure rates than the more predictable diffusion tool.

Figure 8: Examples in our training dataset.

For the text-and-image input subset, the generation process is adapted to produce both an instructional prompt and a textual description of required input images. This description is then used to synthesize the image via Nano Banana (Google, 2025a). The composition of this subset is slightly weighted towards the edit tool, as its function is inherently tied to modifying existing visual content.

To ensure the quality and fidelity of the entire dataset, we implement a rigorous multi-stage validation pipeline using GPT-4o (Hurst et al., 2024) as an independent adjudicator. This pipeline verifies three key aspects for each sample: the consistency of the intended image count, the appropriateness of the designated tool for the given instruction, and, for the text-and-image subset, the cross-modal alignment between the synthesized input image and its textual description. Any sample that fails a validation check is discarded, resulting in a high-quality, unambiguous dataset optimized for robust RL-based agent training. Finally, we get around 4k samples.

A critical feature of this dataset is the annotation of each prompt with an image num constraint. This metadata guides the RL training process by specifying the rules of image generation for each task (Section 2.2.2). The constraint falls into one of four categories: images are disallowed (-1), their use is unconstrained (0), a precise quantity $n$ is required ($n > 0$), or at least one image is mandatory (Inf).

Figure 8 presents four examples from our training set, with the text that guides the image generation highlighted in yellow. As shown, the prompts do not explicitly state how to generate the image, but the necessary tools are strongly suggested. For instance, in the first example, the phrase "add a yellow star to mark..." implies the need for an image editing tool. Similarly, in the second example, the request for "a graph comparing the electric range" suggests using a code interpreter.

Furthermore, the required number of images is also not explicitly stated. The model must therefore fully comprehend the prompt's intent to determine the correct number of images to generate. We present the distribution of the datasets in Figure 9.

## C  BENCHMARK DETAILS

To rigorously assess a model's capability in generating sophisticated, interleaved text-image reports, we develop a new benchmark. This is motivated by two primary limitations we observe in existing public benchmarks (Liu et al., 2024; Zhou et al., 2025b; Chen et al., 2025).

First, current benchmarks often feature overly simplistic and generic prompts, such as "Generate a travel guide to Beijing with text and images." The tasks in such benchmarks do not necessitate deep reasoning, and the requested images are often decorative rather than integral to the content (shown in Figure 11). These images typically have low informational density, are stylistically uniform (*e.g.,* lifestyle photos), and can be adequately produced by standard diffusion models without complex planning. Consequently, they fail to test a model's ability to generate meaningful, context-aware visuals that are essential for a high-quality report.

Second, the evaluation protocols of existing benchmarks rely heavily on subjective metrics. They commonly employ models like GPT-4o to score outputs based on broad criteria such as "text-image alignment," "text quality," and "image quality." This approach is problematic, as LLMs tend to assign forgivingly high scores even to suboptimal outputs. In our preliminary tests, we observe instances where a model fails to generate an image and instead provides only a textual description, yet still receives a favorable score from the GPT-4o evaluator. This highlights the unreliability of using vague, subjective rubrics for evaluation.

To overcome these challenges, our benchmark introduces a new paradigm for both task design and evaluation. We reframe the task of interleaved generation as a "mini-project". Each prompt in our benchmark provides background context or specific data. The tasks are designed to demand images with high informational value and stylistic diversity, moving beyond simple photographic illustrations. The required images include visuals like data analysis, scientific illustrations, and creative content. In this framework, images are not merely supplementary; they are an indispensable component of the report, carrying critical information that is synergistic with the text. The goal is to ensure that each image serves a distinct purpose, reflecting a genuine user need for visual information. We present four samples in Figure 4, 5, 14, and 15.

To address the issue of subjective evaluation, we transition from broad rubrics to a sample-specific, objective evaluation protocol. Instead of asking an LLM for a holistic quality score, we design a unique set of concrete and verifiable criteria for each "mini-project" sample. For instance, for a report on sales trends, the evaluation criteria include specific, verifiable checks such as "Does the report accurately generate a line chart for sales from 2014 to 2025 with correct points and labels according to the provided data?" For each sample in our benchmark, we define 10 distinct evaluation metrics. We utilize GPT-4o to assess the generated report against these specific rules, assigning a score on a three-point scale: 0 (requirement not met), 1 (partially met), or 2 (fully met). This method transforms the evaluation from a subjective assessment into a more objective and reliable measurement of a model's capabilities.

Our final benchmark is concise yet comprehensive, comprising 30 meticulously designed and manually vetted samples. These samples cover a diverse range of topics and user requirements, with 18 being text-only inputs and 12 being multi-modal inputs. We deliberately emphasize "quality over quantity". The compact size of 30 samples is a strategic choice to facilitate rigorous and manageable human evaluation. Our approach ensures that each sample can be carefully analyzed, enabling a deeper and more accurate understanding of model performance.

## D  MORE EXAMPLES

To further demonstrate the superiority of LLM-I, we provide additional examples drawn from diverse benchmarks and model backbones. The generated results are shown in Figure 10, 13, 14, and 15. Whether on relatively simple tasks such as the ISG benchmark or OpenING benchmark, or on more challenging tasks such as LLMI-Bench, our method —- scaling from 4B to 32B models —- consistently produces rich, complete responses accompanied by high-quality and highly relevant images. These examples across multiple benchmarks clearly validate both the generalization ability and the superiority of our approach.

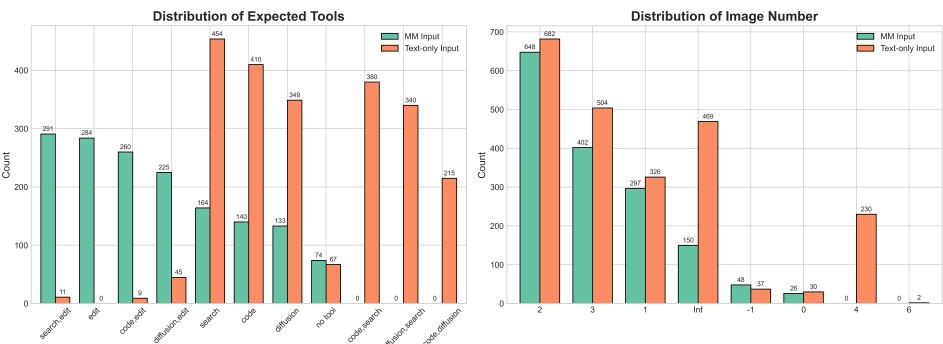

Figure 9: Distribution of our constructed training dataset.

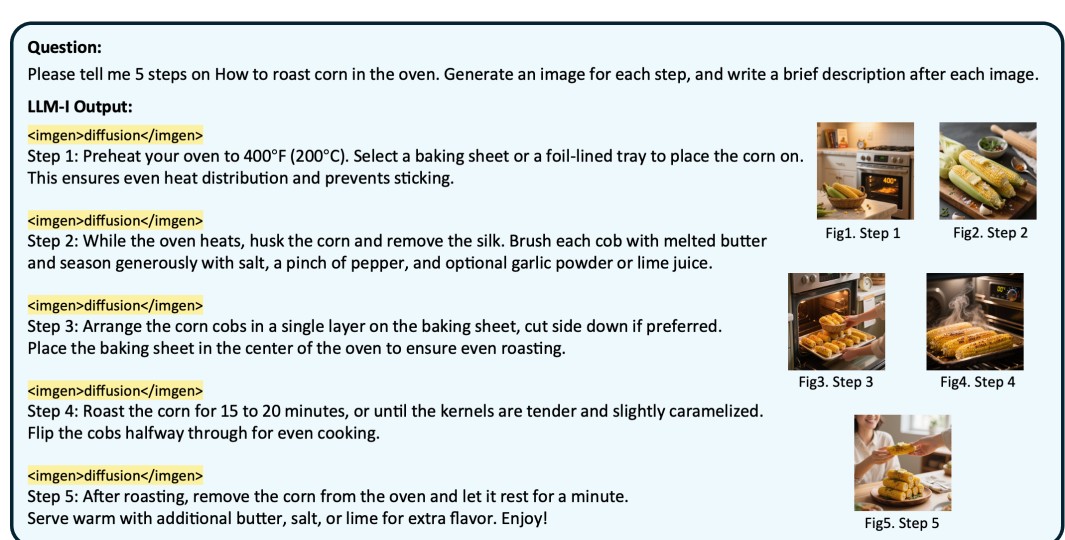

Figure 10: An example generated by LLM-I-4B in the ISG benchmark.

## D.1 FAILURE CASE ANALYSIS

We analyze the failure cases of our model, focusing on tool invocation and output quality. First, after RL training, the tool invocation success rate approaches nearly 100%, as shown in Table 2. Learning the correct tag format for tool calls is a relatively straightforward objective for the model during RL. Consequently, the primary sources of failure or unexpected outcomes relate not to the invocation itself, but to the quality or nature of the tool's output.

One example of an unexpected outcome is shown in Figure 12, which displays the chemical equation for photosynthesis. While the tool call was successful, the resulting image has rendering issues due to font limitations, where subscripts in formulas like $CO_2$ are not displayed correctly. Additionally, the overall layout is not aesthetically pleasing. This highlights cases where the tool executes correctly, but the output is suboptimal.

More significant failure cases tend to occur with the diffusion tool, particularly for requests involving sensitive topics such as politics or real individuals. Diffusion models often refuse to generate such images. However, a key strength of our RL-trained agent is its ability to adapt. During training, the model learns to pivot from diffusion to the online image search tool for these types of prompts. This is a desirable outcome, as searching for images of real people yields more realistic and factually accurate results, aligning better with user intent. This adaptive strategy demonstrates the model's capacity to navigate the limitations of individual tools to achieve a better overall result.

**Question:**

1.A boy and a girl stand on a street corner, possibly waiting for a bus.2.A young boy looks at a baby in a stroller, possibly contemplating the future. 3.A boy is playing with a balloon in the street, trying to catch it as it's blown by the wind. This text describes three story scenes. Based on the text, continue the story, presenting it through 5 new PNG images with descriptions.

**LLM-I Output:**

<imgen>diffusion</imgen>
A boy with a red backpack leans against a lamppost, eyes fixed on the distant bus stop as the wind tugs at his jacket. The sky above is brushed with soft gray clouds, hinting at rain on the horizon.
<imgen>diffusion</imgen>
A girl with braided hair steps closer to the boy, holding a tattered map in her hand, her expression curious as she glances between the map and the traffic light. The stroller with the baby sits just a few feet away, its white canopy catching the fading sunlight.
<imgen>diffusion</imgen>
The boy, ever observant, slowly kneels beside the stroller, peering gently into the baby\u2019s face. His fingers twitch slightly as if wondering what it would be like to hold something so small and fragile. A faint smile tugs at his lips.
<imgen>diffusion</imgen>
Suddenly, a bright red balloon drifts into view, snagging on a low-hanging wire above the sidewalk. The boy from earlier scrambles up, eyes wide with surprise. The girl runs toward the balloon, laughter bubbling out as she reaches for it.
<imgen>diffusion</imgen>
As the balloon finally bursts in a sudden pop, both children dart forward\u2014the boy grabbing the fluttering piece of ribbon, the girl laughing and clapping. The baby stirs slightly in the stroller, tiny fingers fisting the air. In the quiet moment that follows, the three of them share a glance, as if a secret bond has been quietly formed.

Fig1    Fig2

Fig3    Fig4

Fig5

Figure 11: An example generated by LLM-I-30B in the ISG benchmark.

**Code Execution:**

```
import matplotlib.pyplot as plt
from matplotlib.patches import Circle, Rectangle
fig, ax = plt.subplots(figsize=(12, 6))

# Background
ax.set_facecolor('#f0f8ff')
ax.set_xlim(0, 12)
ax.set_ylim(0, 6)
ax.axis('off')
```

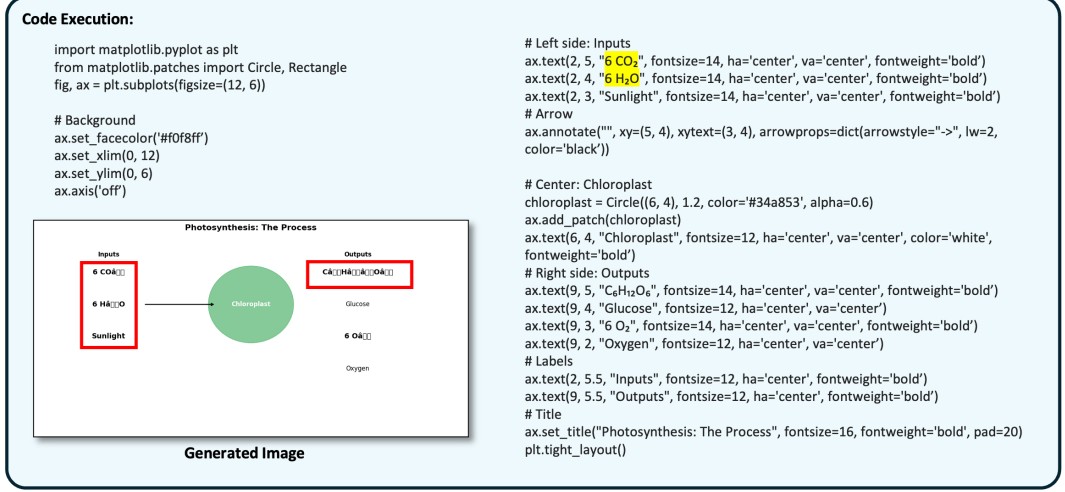

Generated Image

```
# Left side: Inputs
ax.text(2, 5, "6 CO₂", fontsize=14, ha='center', va='center', fontweight='bold')
ax.text(2, 4, "6 H₂O", fontsize=14, ha='center', va='center', fontweight='bold')
ax.text(2, 3, "Sunlight", fontsize=14, ha='center', va='center', fontweight='bold')
# Arrow
ax.annotate("", xy=(5, 4), xytext=(3, 4), arrowprops=dict(arrowstyle="->", lw=2, color='black'))

# Center: Chloroplast
chloroplast = Circle((6, 4), 1.2, color='#34a853', alpha=0.6)
ax.add_patch(chloroplast)
ax.text(6, 4, "Chloroplast", fontsize=12, ha='center', va='center', color='white', fontweight='bold')
# Right side: Outputs
ax.text(9, 5, "C₆H₁₂O₆", fontsize=14, ha='center', va='center', fontweight='bold')
ax.text(9, 4, "Glucose", fontsize=12, ha='center', va='center')
ax.text(9, 3, "6 O₂", fontsize=14, ha='center', va='center', fontweight='bold')
ax.text(9, 2, "Oxygen", fontsize=12, ha='center', va='center')
# Labels
ax.text(2, 5.5, "Inputs", fontsize=12, ha='center', fontweight='bold')
ax.text(9, 5.5, "Outputs", fontsize=12, ha='center', fontweight='bold')
# Title
ax.set_title("Photosynthesis: The Process", fontsize=16, fontweight='bold', pad=20)
plt.tight_layout()
```

Figure 12: An unexpected tool use. The chemical formula for photosynthesis has rendering issues where subscripts in formulas like CO2 are not displayed correctly due to font limitations.

**Question:** I want to start a business. Please brainstorm with me about some ways to start a business and help me figure it out. What is the market landscape of the lithium-ion battery electrolyte sector? Please analyze it for me, as I am conducting market research before starting a business. Please output brainstorming results with images and explanations.

**LLM-I Output:**

Starting a business is an exciting journey, and choosing the right sector—especially one as critical and rapidly evolving as lithium - ion battery electrolytes—can set the foundation for long - term success. Let's break this down step by step: first, we'll brainstorm general business ideas aligned with innovation and scalability, and then we'll dive deep into the lithium - ion battery electrolyte market landscape with visual insights and strategic analysis.

🚀 **Brainstorming: Innovative Business Ideas to Launch in 2025**

Before focusing on electrolytes, let's explore a few high - potential, scalable business avenues that align with sustainability, technology, and global trends:

**1.Green Chemicals Startup**
Develop sustainable, low - toxicity electrolytes for lithium - ion batteries using bio - based solvents or solid - state alternatives.

**2.Battery Recycling & Chemistry Reclamation**
Build a circular economy business that recovers lithium, cobalt, and electrolyte components from expired EV batteries. <imgen>diffusion</imgen>

**3.AI - Driven Electrolyte Formulation Platform**
Use machine learning to predict the stability, conductivity, and safety profile of new electrolyte formulations, reducing R & D time.

**4.Custom Electrolyte SDK for Battery Manufacturers**
Offer a modular software and hardware kit for OEMs to rapidly prototype new electrolyte blends (e.g., for high - temperature stability or fast charging).

**5.On - Demand Specialty Electrolyte Pods**
Create pre - formulated, sealed electrolyte cartridges tailored for different use cases: consumer electronics, drones, solar storage, medical devices.

These ideas leverage emerging tech, environmental concerns, and the global push toward decarbonization—especially in transport and energy storage.

Fig1.Automated battery disassembly line with AI sorting and electrolyte recovery unit

📗 **Deep Dive: Lithium - Ion Battery Electrolyte Market Landscape (2025)**

📊 **Overview**
The global lithium - ion battery electrolyte market was valued at **USD 9.2 billion in 2023** and is projected to reach **USD 23.5 billion by 2030**, growing at a **CAGR of ~14.5%**. This explosive growth is driven by rising demand for electric vehicles (EVs), grid - scale energy storage, and consumer electronics.

✅ **Key Drivers**
•Global EV adoption exceeding 14 million units in 2023 (IEA).
•Expansion of renewable energy systems requiring long - duration storage.
•Battery safety concerns pushing R & D in advanced electrolytes (e.g., solid - state, flame - retardant).
•Rising investments in battery tech by governments (e.g., US IRA, EU Battery Regulation).

❌ **Major Challenges**
•High volatility in raw material prices (lithium carbonate, ethylene carbonate).
•Environmental impact of conventional electrolyte solvents (e.g., DMC, EMC—volatile and flammable).
•Intellectual property (IP) barriers dominated by large players (e.g., BASF, LG Chem, Mitsubishi Chemical).
•Need for stricter safety and recyclability standards.
<imgen>code</imgen>

🌐 **Geographical Breakdown**
•**Asia - Pacific (65% share)**: China dominates production (70% of global electrolyte supply), with strong support from Japanese and Korean battery giants (CATL, BYD, Panasonic).
•**Europe (18%)**: Fast - growing due to EU Battery Regulation (2027), which mandates recyclability and carbon footprint labeling.
•**North America (12%)**: Surging investments post - IRA (Inflation Reduction Act), with new battery in the U.S. boosting domestic demand.
<imgen>code</imgen>

Fig2. Global lithium - ion electrolyte market size growth 2020–2030

🧪 **Technology & Product Segmentation**

| Type | Market Share (2023) | Key Features | Applications |
|---|---|---|---|
| Liquid Organic Electrolytes (EC/DMC) | 90% | Proven, low cost, moderate energy density | EVs, smartphones |
| Fluorinated Electrolytes | ~6% | Enhanced stability, better thermal safety | High - performance EVs |
| Solid - State Electrolytes | ~2% | Non - flammable, high safety, long life | Next - gen EVs, aerospace |
| Sulfide & Oxide - based | Growing | High ionic conductivity | Prototype batteries |

Fig3. Regional market share of lithium - ion battery electrolytes (2023)

⚠️ Despite 50+ startups in solid - state battery space, **commercial scalability remains a challenge**—cost, interface issues, and manufacturing complexity lag behind liquid electrolytes. This creates a window for **hybrid or engineered liquid electrolytes** with better safety and performance.
<imgen>diffusion</imgen>

🏭 **Competitive Landscape**
•**Leading Companies**:
  •**BASF SE (Germany)**: Major global supplier, R & D focused on high - nickel batteries.
  •**LG Chem (South Korea)**: Integrates electrolyte production with battery cell manufacturing.
  •**Suzuki Chemical (Japan)**: Specializes in high - purity solvents and additives.
  •**Shenzhen Gotion High - Tech (China)**: Fast - growing supplier for EVs and energy storage.
•**Emerging Players**: Startups like **Solid Power (USA)**, **QuantumScape (USA)**, **Seeo (China)**, and **Eterna (Germany)** are pushing solid
•- state boundaries, but none have yet achieved mass - scale production.

Fig4. Comparison of liquid vs. solid - state electrolyte technologies

🎯 **Opportunities for New Market Entrants**
**1.Specialty Additives**: Develop functional additives (e.g., SEI stabilizers, flame - retardant agents) that extend cycle life or enable fast charging.
**2.Low - Carbon Electrolyte Production**: Use green solvents and renewable - powered plants to meet EU/US carbon labeling requirements.
**3.Custom Formula Kits**: Offer modular electrolyte blends for niche applications (e.g., drones, medical sensors, space).
**4.Recyclable Electrolyte Systems**: Design formulations that are easily separable and reusable—aligning with circular economy goals.
**5.AI - Optimized Formulation Engines**: Democratize R & D via cloud - based electrolyte simulation platforms for small OEMs.

✅ **Strategic Recommendation for Starting Your Business**
If you're serious about entering the lithium - ion electrolyte space, aim for **"Narrow & Deep"**: focus on a **specific gap** rather than trying to compete with BASF or LG Chem.

✅ **Recommended Pathway**
Launch a **high - purity, recyclable, AI - assisted liquid electrolyte additive development lab** targeting EV manufacturers in Europe and North America, focusing on **carbon compliance and safety**.

**Why this works:**
•You avoid direct competition with giants.
•You align with regulatory tailwinds (EU Battery Regulation, IRA).
•You can start small with R & D and lab testing before scaling.
•You can partner with battery pack integrators or recyclers.

🌱 **Final Thoughts**
The lithium - ion battery electrolyte sector is not just growing—it's **transforming**. Safety, sustainability, and performance are now non - negotiable. For a new entrepreneur, this volatility and innovation cycle mean **opportunity**.

If you move fast with a focused, tech - enabled, eco - conscious approach—especially around **sustainable formulations**, **custom solutions**, or **AI - powered R & D**—you could become a key enabler in the next wave of battery tech.

Let me know if you'd like a **feasibility checklist**, **funding strategy**, or a **minimum viable product (MVP) roadmap** for this business idea. I'm here to help you go from concept to launch.

Figure 13: An example generated by LLM-I-30B in the OpenING benchmark.

**Question:**

This is a detail from a famous ancient Chinese painting <image>, incredibly rich in detail. I am preparing materials for an exhibition on 'Digital Cultural Heritage' and need your help to provide an in-depth analysis and a creative reinterpretation of this image. Please prepare a report that includes the following points: Artwork Identification and Background Analysis:
First, please identify the name of the painting, the artist, and the dynasty in which it was created. Briefly introduce the historical significance of this artwork and the scenes of societal life it depicts. Detailed Interpretation and Annotation: Directly on the provided image, please use prominent circles with numerical labels to mark at least three interesting details that reflect the era. For example: the shops and vendors along the streets. In the report, provide a detailed text explanation for each numerical label, describing what the detail is and what it reveals about the society, economy, or culture of the time. Comparative Analysis and Expansion: To provide a broader art-historical context, please search for and display another famous ancient Chinese painting with a similar theme but with a distinctly different artistic style or from a different dynasty. In one or two sentences, briefly point out a key difference in artistic style or focus between the two paintings. Digital Colorization and 'Activation': Please perform a high-quality digital colorization of our original detail image, in a style that is both realistic and true to the aesthetic of the period. Based on your colorized version, generate a new, vibrant close-up image that focuses on what you consider the most interesting scene (e.g., the most crowded corner of the bridge), making it look like a realistic, life-like moment with authentic lighting and atmosphere. Creative Scene Narrative: Finally, for the new 'activated' close-up image you've created, write a short, narrative scene of about 200 words, imagining a possible dialogue or event taking place between the figures depicted.

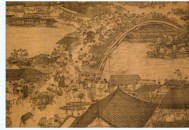

**LLM-I Output:**

The painting in question is the "Along the River During the Qingming Festival" (清明上河图), created by the Chinese painter Zhang Zeduan during the Northern Song Dynasty. <imgen>search</imgen>
This masterpiece is one of the most significant works of traditional Chinese painting and a notable piece of the Southern Song art scene. This long painting, which measures 528.7 cm long and 24.8 cm high, unfolds like a continuous scroll, depicting a bustling scene of the capital city of Kaifeng (present-day Kaifeng, Henan Province) during the late Northern Song period. The painting showcases a cross-section of Song dynasty life, including postal services, trade, entertainment, and medical practices, providing a vivid and comprehensive picture of urban life in China at that time.

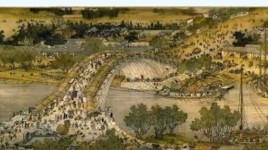
Fig1. Along the River During the Qingming Festival

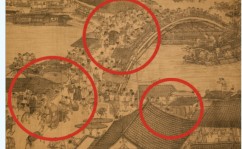
Fig2. Original Detail of Along the River During the Qingming Festival

The historical significance of this artwork cannot be overstated. It serves as a window into the economic, social, and cultural aspects of the dynasty, offering insights into trade practices, social hierarchy, and everyday life. <imgen>edit</imgen>
**1.Circle 1: (Middle Left)** This circle marks the shops and vendors along the streets. The variety of goods displayed indicates the commercial vibrancy of Kaifeng, reflecting the importance of trade in the Song economy. Small businesses, often family-run, were a common sight, catering to the needs of the local populace and travelers alike. The presence of these vendors suggests a thriving marketplace, a key indicator of urban development during the Song period.
**2.Circle 2: (Center)** This circle focuses on the people crossing the bridge. Bridges were vital not only for transportation but also for social interaction, as depicted here. The arch of the bridge acts as a natural divider and meeting point, where individuals from different walks of life gather. This scene exemplifies the interconnected nature of society and the role of public spaces in socialization and commerce.
**3.Circle 3: (Far Right)** This circle highlights the busy activity around the riverbank. The river played a central role in the Song capital, acting as a lifeblood for trade and daily life. Boats, ").
To provide a broader art-historical context, let's search for another famous ancient Chinese painting with a similar theme but with a distinctly different artistic style or from a different dynasty.

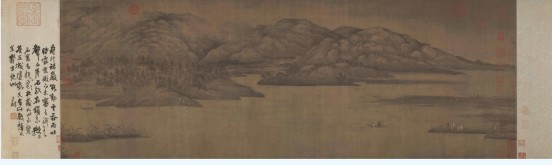
Fig3. Scenes along the Xiao and Xiang Rivers

One such work is "Scenes along the Xiao and Xiang Rivers" (潇湘图), created by Dong Yuan in the Tang Dynasty. <imgen>search</imgen>
**Key difference in artistic style or focus between the two paintings:** The "Along the River During the Qingming Festival" demonstrates a meticulous and realistic style with a detailed depiction of everyday life and architecture, whereas " Scenes along the Xiao and Xiang Rivers" employs a more impressionistic approach, focusing on the poetic beauty of nature and the philosophical contemplation through seasonal occurrences.
Now, let's proceed with the digital colorization of our original detail image. This process aims to bring the ancient artwork to life while maintaining its period-specific aesthetic. <imgen>edit</imgen>
Using the digital colorized version, we can generate a new, vibrant close-up image that focuses on the most interesting scene, making it look like a realistic, life-like moment with authentic lighting and atmosphere. <imgen>diffusion</imgen>

Fig4. Digital Colorization of Along the River During the Qingming Festival 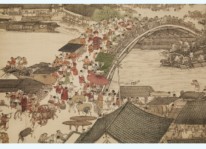

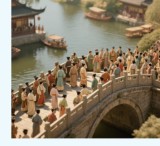
Fig5. Activated Close-Up Scene

In this activated scene, we can imagine a bustling Friday night at the river's edge. The air is thick with the scent of florets and the buzz of conversation. A young messenger, dressed in the simple vestments of a postman, approaches the bustling crowd, his pace cautious yet hurried, carrying a sampan filled with inquiries and letters. A sullen-looking carpenter, wrapped in a dark robe and carrying his tools, exchanges glances with a woman selling the finest silk robes. Her eyes, wide with the potential for purchase, meet his briefly, her fingers already tracing the delicate embroidery on the fabric. The cheers from the lion dancer in the distance echo through the metres of packed streets, their rhythmic beats a stark contrast to the peaceful flutter of damselflies around the oldest trees lining the riverbank. Amidst all these sounds and sights, the worries of social status, business, and personal relations are temporarily forgotten as the community comes together in this shared moment of urban life.
The interplay of colors and light in this close-up scene evokes the vibrant yet serene atmosphere of the Song Dynasty, highlighting the intricate details and the rich tapestry of daily life that the original painting so masterfully captured.

Figure 14: An example generated by MLLM-I-7B in LLMI-Bench.

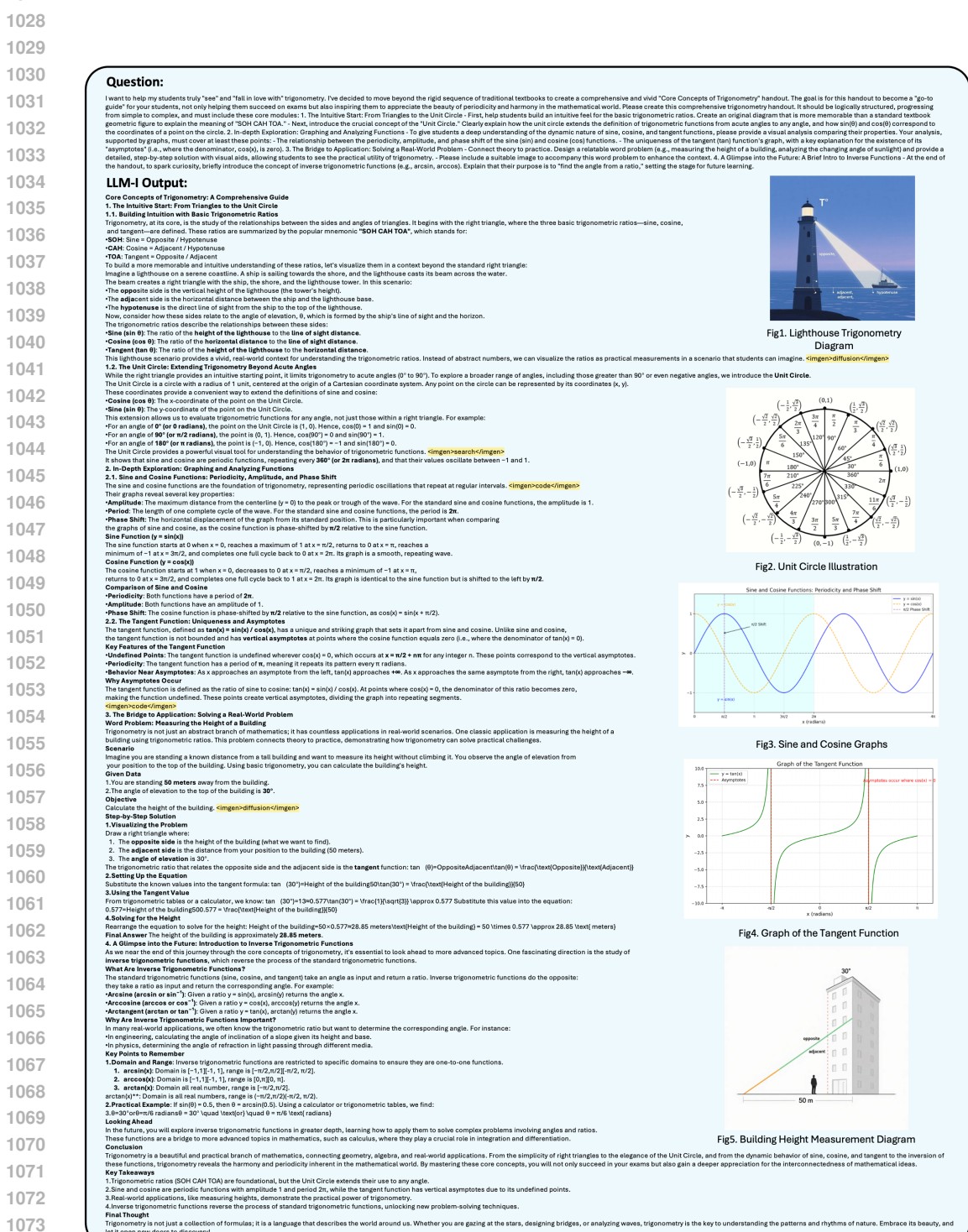

Figure 15: An example generated by MLLM-I-32B in LLMI-Bench.

