# OpenReview forum: "LLM-I: LLMs are Naturally Interleaved Multimodal Creators"
_ICLR.cc/2026/Conference — Submitted to ICLR 2026_

### Official Review · Reviewer_25UT · 2025-10-26

**Soundness:** 3
**Presentation:** 2
**Contribution:** 2
**Rating:** 4
**Confidence:** 4

**Summary:**

This paper introduces LLM-Interleaved, a framework that reframes interleaved image-text generation as a tool-use problem, empowering an LLM/MLLM agent to orchestrate diverse visual tools (e.g., search, diffusion, code) to overcome factual grounding and precision limitations. Trained via a Reinforcement Learning framework with a hybrid reward system, LLM-I achieves state-of-the-art performance across four benchmarks and is further enhanced by a novel test-time scaling strategy.

**Strengths:**

Current unified models are often limited to synthetic imagery and struggle with tasks requiring factual grounding or programmatic precision. LLM-I addresses these limitations by enabling the LLM agent to intelligently select and apply tools like online image search, diffusion-based generation, code execution, and image editing. This allows for the creation of more accurate and diverse visual content.

LLM-I reframes interleaved image-text generation as a tool-use problem, allowing a central Large Language Model (LLM) or Multimodal Large Language Model (MLLM) to orchestrate a diverse toolkit of specialized visual tools. This modular approach is more flexible than unified models, which often suffer from a "one-tool" bottleneck.

**Weaknesses:**

1. Limited Benchmark Size: The newly proposed LLMI-Benchmark consists of only 30 samples. This small dataset size is insufficient to draw convincing conclusions about the model's performance on this specific benchmark.

2. Marginal Performance Gains: The performance advantage of LLM-I is not consistently significant across Tables 1, 2, and 3. For instance, in Table 1, the GPT-4o + DALL-E 3 baseline shows competitive results, outperforming MLLM-I-7B and -32B. Similarly, ISG also demonstrates strong performance on image and structural metrics, surpassing the MLLM-I variants.

3. The proposed test-time scaling strategy appears to be computationally intensive. However, the paper only offers a theoretical analysis without quantitative experimental results (e.g., latency, resource usage) to assess its practical overhead.

Minor Typos:
Line 082: "Thourgh" should be "through".
Caption of Table 2: "LLM-Bench" should likely be "LLMI-Bench".

**Questions:**

The results surprisingly indicate that LLM-I variants often outperform MLLM-I variants (e.g., 4B LLM-I is nearly on par with 32B MLLM-I). Could the authors provide an analysis or hypothesis for this phenomenon?

Tool call checking seems critical for generation quality, yet it is only mentioned as part of the test-time scaling strategy. Why was this verification step not included as a standard part of the core LLM-I framework?

The authors state that existing benchmarks "do not necessitate deep reasoning." Could the authors elaborate on why "deep reasoning" is a crucial aspect to emphasize specifically for interleaved generation tasks? Besides, regarding the cooking example in Figure 10 (HowTo task), images are necessary and instructional, not decorative as the authors say. Images in HowTo task can effectively guide a user who is unfamiliar with the process like cooking. I suggest using a more representative example to state the issue.

---

> ### Author Response · Authors · 2025-11-12
>
> Thank you for your constructive feedback. We address the weaknesses and your question below.
>
> **Q1: Benchmark size.**
>
> **R1:** The LLMI-Bench is intentionally small to accommodate rigorous human evaluation and a detailed, rubric-based scoring protocol, which is not feasible for benchmarks with thousands of automatically generated samples. We believe this focus on quality over quantity provides a more reliable assessment of complex reasoning. To ensure a comprehensive evaluation, we have also benchmarked our model on three other large-scale benchmarks (OpenING, ISG), which validate the generalizability of our method.
>
> **Q2: Performance gains, especially in comparison to the GPT-4o + DALL-E 3 baseline.**
>
> **R2:** This is a great point. There are two key reasons for the strong performance of the GPT-4o baseline on certain benchmarks:
>
> 1.  **Evaluator Bias:** As noted in the OpenING paper itself, GPT-4o (which is used as the evaluator) tends to prefer outputs from OpenAI models.
> 2.  **Pipeline Simplicity vs. Flexibility:** The GPT-4o + DALL-E 3 method is a simple, two-stage pipeline: first generate all text, then generate images. This pipeline is inflexible (e.g., no control over image count or placement) but is also less prone to certain types of errors. Since the OpenING benchmark does not strictly enforce constraints on image count, this simpler approach can achieve high scores.
>
> Qualitatively, we observe that the outputs from MLLM-I are far superior in terms of coherence and contextual appropriateness. Our results are based on the official, standard evaluation protocols for each benchmark. A similar point applies to the ISG benchmark, where the "structural metric" specifically measures image position. Since the ISG model was designed by the creators of the ISGBench, it is essentially an in-domain evaluation for them, and their model is highly optimized for that specific metric.
>
> **Q3: Quantitative computational resource results for TTS.**
>
> **R3:** We have included these results in the paper. As shown in **Table 5**, a sample processed with our full TTS strategy incurs an additional latency of approximately **~20 seconds**.
>
> **Q4: Why LLM-I variants often outperform MLLM-I variants.**
>
> **R4:** This is an excellent observation. The performance difference stems primarily from the underlying base models:
> -   The backbone for **LLM-I is the Qwen3 series**.
> -   The backbone for **MLLM-I is the Qwen2.5-VL series**.
>
> The Qwen3 models are significantly more capable than the Qwen2.5 models in core language tasks like creative writing and complex instruction following. Since our framework offloads visual generation to external tools, the model's ability to plan and write high-quality text is paramount, giving the stronger LLM backbone an advantage.
>
> **Q5: Why tool call checking was not a standard part of the core framework outside of TTS.**
>
> **R5:** A lightweight version of tool call checking *is* part of the core framework. During RL training, the rule-based reward component penalizes malformed tags. At inference, a basic check ensures structural validity.
>
> The "Tool Call Check" in TTS is more comprehensive, as it is part of a multi-stage process that includes a **selector** (which functions like a Best-of-N filter). As we show in Table 2, the tool invocation success rate is nearly 100% after RL training, so a simple check alone provides only marginal gains. The more advanced check in TTS is designed to prevent rare failures before proceeding to the more resource-intensive selection and enhancement stages.
>
> **Q6: Elaboration on "deep reasoning" for interleaved generation and the choice of example.**
>
> **R6:** We are happy to elaborate. A key application for interleaved generation is the creation of complex reports, such as market or data analyses. This requires the model to analyze data, formulate a coherent narrative structure, and strategically reason about where and how to present images to convey information most effectively. The model must synthesize information and decide which data is best represented visually. This process is analogous to a mini-project that, in a corporate setting, would require an experienced expert to manage.
>
> Regarding the cooking example, we appreciate you pointing out that "how-to" images are instructional, not merely decorative. You raise an excellent point. The distinction we aim to highlight is between **generic, illustrative images** and **high-information, synergistic visuals** that are deeply integrated with the text. And we will include such an example for better illustration of decorative images in our manuscript.

---

> ### Author Response · Authors · 2025-11-21
> **Manuscript Update on "Decorative" vs. "Instructional" Images and Typos**
>
> Dear Reviewer 25UT,
>
> We appreciate your insightful comment regarding "HowTo" images (Q6). To better clarify our definition of "decorative images" with low information gain, we have updated the manuscript with a more representative example in **Figure 11**. This sample features a story about a boy with a red backpack and people releasing balloons. In this case, the images are purely ornamental: the text "a little boy with a red backpack" already conveys all necessary information, and the corresponding image provides no additional information gain or necessary visual context. By contrast, the tasks in our benchmark (like the data analysis example discussed in Appendix B) require images that are indispensable for understanding the content. We hope this new example and explanation clarify our perspective. Besides, we have revised the typos in line 82 and table 2. Thank you for your reminder.

---

### Official Review · Reviewer_ByY3 · 2025-10-27

**Soundness:** 2
**Presentation:** 3
**Contribution:** 1
**Rating:** 2
**Confidence:** 3

**Summary:**

This paper introduce the LLM-I framework, which restructures image–text interleaved generation through “tool scheduling.” A central LLM/MLLM orchestrates four complementary tools: online image retrieval, diffusion-based generation, code-driven visualization, and image editing. It is trained with reinforcement learning that combines rule-based and LLM/MLLM-evaluated hybrid rewards.

**Strengths:**

* The integration of four complementary tools covers a wide spectrum of use cases: real photographs are obtained via search, data charts are produced by executing code, and creative visuals are synthesized through diffusion models.

* A dedicated dataset of 4,000 samples was constructed in which the model is required to “select tools implicitly” (i.e., without explicitly naming them), thereby strengthening its reasoning capabilities.

**Weaknesses:**

* Tool scheduling  and test-time scaling markedly increase latency, making the framework ill-suited for real-time, low-latency interactions.

* There is no diagnostic analysis of tool failures, such as code-execution errors, or editing results that fall short of expectations, so it remains unclear which tools are the weak links.

**Questions:**

* The reinforcement learning reward only judges the tool selection and output matching, while ignoring the limitations of the tool (e.g., low-resolution search results, diffusion distortion). As a result, the correct choice is penalized due to the tool's shortcomings, thus breeding error aversion. How to solve this problem?

---

> ### Author Response · Authors · 2025-11-12
>
> Thank you for your constructive feedback. We address the weaknesses and your question below.
>
> **Q1: Latency of tool scheduling and test-time scaling.**
>
> **R1:** It is true that agentic systems often trade latency for improved accuracy and comprehensiveness. For example, popular web-enabled assistants like ChatGPT and Gemini can take significant time (from many seconds to minutes) to search and reason before providing a reliable answer.
>
> Our framework offers two deployment modes to manage this trade-off:
> -   **Real-time mode:** Tool calls are executed in parallel, so latency is determined by the slowest tool. This mode is designed for low-latency interactions.
> -   **Quality mode:** This mode uses our full test-time scaling (TTS) pipeline for maximal quality. As reported, the end-to-end overhead is a practical ~20 seconds per item.
>
> Compared to typical web-browsing agent latencies, our real-time mode is substantially faster. Our quality mode also remains practical, largely due to our parallel and asynchronous implementation. For a typical task, our system can generate a vivid report with informative and instructional images in under a minute.
>
> **Q2: Analysis of tool failures.**
>
> **R2:** First, we wish to clarify that after RL training, the **tool invocation success rate approaches nearly 100%** (as shown in Table 2). Learning the correct tag format for tool calls is a relatively straightforward objective for the model during RL. The more nuanced failures relate to the *quality* of the tool's output (e.g., a blurry searched image), which we address in the next point. We will also add some failure or unexpected examples to our manuscript. Thank you for your advice.
>
> **Q3: How the reward function handles inherent tool limitations (e.g., low-resolution search results) to avoid penalizing a correct tool choice.**
>
> **R3:** This is an excellent question. Our framework mitigates this issue through both the RL design and engineering safeguards.
>
> From an RL perspective:
> -   Our **LLM judge directly evaluates the quality of the tool selection and its parameters** (e.g., the "tool source" and the text prompt). This ensures the model receives credit for making the right choice, even if the external tool's execution is noisy.
> -   The learning process is robust to occasional low-quality outputs. The final gradient is an average over a batch of rollouts, so a small fraction of noisy results does not alter the overall learning trajectory. As shown in Figure 6, the **Tool F1 score (precision and recall) steadily increases** during RL, indicating the model is not developing tool aversion.
> -   For many tasks (e.g., plotting a sine wave), **multiple tools are viable** (code execution or image search). The RL process naturally steers the agent toward the tool that most reliably produces a high-quality result, building in resilience.
>
> From an engineering perspective:
> -   **Search:** Our tool, using the Google API, returns N results per query. Before these are used, we apply **automated, rule-based filtering** for resolution and other quality heuristics. During TTS, an MLLM selects the best-aligned image from multiple candidates, further buffering against noise.
> -   **Diffusion:** While we have not observed consistent distortion issues, our framework can be configured to generate multiple candidates in parallel and select the best one.

---

> ### Author Response · Authors · 2025-11-21
> **Manuscript Update on Failure Case Analysis**
>
> Dear Reviewer ByY3,
>
> Following up on our previous response regarding tool failures (Q2), we have now updated our manuscript to include a dedicated **"Failure Case Analysis"** section in **Appendix**.
>
> This new section provides a detailed discussion on the nature of tool failures, distinguishing between invocation errors (which are rare after RL) and output quality issues. We have also added **Figure 12**, which showcases a specific example of a suboptimal tool output (rendering issues in a chemical formula). Additionally, we discuss how our agent learns to adaptively switch tools (e.g., from diffusion to search) when facing refusals on sensitive topics. We hope this addition addresses your request for a concrete analysis of failure modes.

---

### Official Review · Reviewer_4c1Z · 2025-10-27

**Soundness:** 2
**Presentation:** 2
**Contribution:** 2
**Rating:** 4
**Confidence:** 2

**Summary:**

The paper introduces LLM-Interleaved (LLM-I), a tool-augmented agent for interleaved text–image generation. A central LLM/MLLM is fine-tuned with RL on a tool-oriented synthetic dataset using a hybrid reward—rule-based checks for image count and tag format, plus LLM/MLLM judges. At inference, a multi-stage test-time scaling pipeline performs candidate sampling, tool-call validation, selection, tool-specific enhancement, and polishing. Evaluations on OpenING, ISG, an in-domain set, and the new LLMI-Bench show strong results; LLMI-Bench additionally includes human ratings and a tool-invocation accuracy metric.

**Strengths:**

1.Clear tool-calling protocol with a structured placeholder tag enabling single-pass orchestration.

2.Explicit hybrid reward with rule-based gating; equations and components are well documented.

3.TTS pipeline is described step-by-step and shows stagewise gains.

4.Competitive results across OpenING/ISG/LLMI-Bench, with tool and reward ablations.

**Weaknesses:**

1.Limited novelty: The contribution reads as a recipe (tool use + hybrid rewards + TTS) rather than a new learning principle, integrating known ideas: inline tool tags, judge-based RL with rule constraints, and TTS reranking/polishing.

2.Evaluation setup: The in-domain evaluation reuses training-time metrics (rule/LLM/MLLM judges).

3.Metric interpretation: “Tool Acc = 100” reflects invocation success, not task correctness.

4.Data scale: The RL dataset is small (~4k) and LLMI-Bench is tiny (30 items).

**Questions:**

None.

---

> ### Author Response · Authors · 2025-11-12
>
> Thank you for your thoughtful comments. We address each weakness concisely.
>
> **Q1: Limited novelty: The contribution reads as a recipe (tool use + hybrid rewards + TTS) rather than a new learning principle.**
>
> **R1:** The core novelty of our work is a new paradigm for interleaved image-text generation. Unlike existing unified or two-stage methods, we reframe the task as a **single-pass, agentic orchestration of a heterogeneous visual toolkit** managed via a typed inline tag. This approach directly addresses the critical "one-tool" limitation of current methods. The RL framework and test-time scaling are the mechanisms we designed to make this new paradigm effective and robust. This is a practical, end-to-end recipe tailored specifically for this task, which prior tool-use agents have not targeted.
>
> **Q2: Evaluation setup: The in-domain evaluation reuses training-time metrics.**
>
> **R2:** Thank you for pointing this out. We wish to clarify that while the *metrics* are the same, the *judge* is different. For our in-domain evaluation, the training-time judges are Qwen-based, whereas the test-set evaluation uses **GPT-4o**. Using an independent, powerful model as the judge for evaluation demonstrates that our model's performance is not due to overfitting to the specific biases of the training judges. This, combined with our strong performance on three OOD benchmarks (which include model-based, rule-based, and human evaluations), makes our evaluation framework robust.
>
> **Q3: Metric Interpretation of "Tool Accuracy".**
>
> **R3:** For an open-ended task like interleaved generation, a single "correctness" score is ill-defined. Consequently, prior work relies on multi-faceted evaluations of image and text quality. In this context, **tool invocation success is a critical and necessary proxy for task completeness**. A failed tool call results in a missing image, which directly harms the integrity of the final multimodal sequence. Furthermore, tracking this metric quantifies the agent's competence before and after RL, clearly demonstrating the effectiveness of our training regimen.
>
> **Q4: Dataset scale.**
>
> **R4:** The RL training set is intentionally curated to be high-signal for tool use (featuring implicit constraints and a diversity of challenging tools). We found this was sufficient for the model to learn the task effectively, and its ability to generalize is evidenced by its strong performance on large OOD benchmarks (OpenING/ISG) and independent human/rubric scores.
>
> A dataset of ~4k samples is common for RL-based agent tuning. Numerous recent studies show that for agent training, **data quality is often more critical than sheer quantity**. For instance, the recent paper "Less is More for Agency" demonstrates that with only 78 carefully designed training samples, their model achieved 73.5% on comprehensive agency benchmarks, underscoring that a smaller, high-quality dataset can be highly effective.
>
> Regarding the LLMI-Bench (30 samples), its smaller size is a deliberate choice to enable rigorous human evaluation and detailed, rubric-based scoring, which is not feasible at the scale of thousands of auto-generated samples. This reasoning-heavy benchmark, combined with the three other large-scale benchmarks, provides a comprehensive validation of LLM-I's capabilities and generalizability.

---

### Official Review · Reviewer_QUx9 · 2025-10-30

**Soundness:** 3
**Presentation:** 3
**Contribution:** 4
**Rating:** 8
**Confidence:** 4

**Summary:**

This paper proposes a framework called LLM-Interleaved (LLM-I), which uses Reinforcement Learning (RL) to train an LLM/MLLM agent, enabling it to intelligently 'orchestrate' and use multiple visual tools (like image search, generation, editing, etc.) to solve complex interleaved image-text generation tasks.

**Strengths:**

## Strengths:

- This paper describes a framework for building and training a multimodal agent, centered around an LLM/VLM, which orchestrates vision tools to achieve interleaved image-text generation. I believe this is a valuable exploration for expanding the boundaries of agent capabilities. I am very optimistic about the direction of multimodal agents, and this work represents a solid exploration in this area.

- The experiments are comprehensive, incorporating both model-based scores and rigorous human evaluations. LLM-I demonstrates superior performance on both in-domain and OOD benchmarks.

**Weaknesses:**

## Weaknesses:

- I have concerns regarding the choice of the reward signal. Using two large-scale LLMs/VLMs as scorers during the RL process incurs significant computational overhead. The authors should provide a runtime analysis, including the number of GPUs used and the total execution time.

- Generative reward models are susceptible to "reward hacking." Although the performance results appear fine, I would like to know if the authors considered this risk when designing the LLM-I system.

- If I recall correctly, the OpenING dataset contains 5,400 samples. Why does Line 288 state that there are only 2,000 samples?

Apart from these points, I have no other major concerns. Overall, this is an excellent piece of work.

**Questions:**

See weakness.

**Details Of Ethics Concerns:**

Nan

---

> ### Author Response · Authors · 2025-11-12
>
> Thank you for your thoughtful and encouraging review. We address your concerns below.
>
> **Q1: Computational overhead of using two large-scale LLMs/VLMs as scorers.**
>
> **R1:** We agree that a clear runtime accounting is important. We have recorded the throughput and timing details during RL training. Here is a summary for the LLM-I-4B and LLM-I-30B models:
>
> | Model         | Time of LLM Judge (per sample) | Time of MLLM Judge (per sample) | Time per Step (BS=32) | Throughput (sequences/sec) |
> |---------------|--------------------------------|---------------------------------|-----------------------|-----------------------------|
> | **LLM-I-4B**  | < 1s                           | 5-7s                            | ~500s                 | ~38                         |
> | **LLM-I-30B** | < 1s                           | 5-7s                            | ~1200s                | ~18                         |
>
> *Time per Step* includes rollout, reward calculation, and the backward update. Although our LLM judge (Qwen3-235B) is large, it is a Mixture-of-Experts (MoE) architecture with only 22B activated parameters per forward pass, and it does not process images, making it very fast. The two judges need 8*80GB GPU memory for deployment.
>
> To further accelerate training, we implemented a parallelized framework where reward calculation and tool invocation are batched and executed asynchronously. This allows LLM/MLLM judges and the various tool calls (search, diffusion, code, edit) to run in parallel, significantly improving efficiency.
>
> **Q2: Reward hacking risk with generative reward models.**
>
> **R2:** We agree this is a critical concern. In the RL design stage, we have considered this issue. To mitigate this risk, we therefore designed our reward function with the following safeguards:
>
> -   **Hybrid Reward with Gating:** A deterministic, rule-based component enforces structural constraints (e.g., image count). Crucially, the MLLM reward is gated by this rule-based score, meaning the model cannot gain points for visual quality if it fails to adhere to the basic structural requirements (like omitting a required image).
> -   **Complementary Judges (Most important):** We use two distinct judges—a text-centric LLM and an image-text MLLM. While their functions overlap, they have different focuses. For instance, the LLM judge evaluates the quality of the generated *tool-use tag* (e.g., the text prompt for the diffusion model), while the MLLM judge evaluates the final *rendered image*. **This design prevents the agent from exploiting the blind spots of a single reward model.**
> -   **Strategic Dataset Construction:** The training dataset explicitly over-represents tasks requiring more challenging tools (code, search) to discourage the model from developing "reward-avoidance" behaviors where it defaults to easier tools.
>
> Furthermore, our evaluation on the in-domain test set (Table 4) uses the same metrics but substitutes a different model (GPT-4o) as the judge. The strong performance gains observed with this independent judge indicate that our model has not overfit to the specific biases of our training judges, providing robust evidence against reward hacking.
>
> **Q3: Sample number in the OpenING benchmark.**
>
> **R3:** Thank you for the question, and we apologize for the confusion caused in the manuscript. The reference to "over 2000 samples" was intended to describe the text-only input sample in OpenING for LLM-I. The full **OpenING benchmark**, which was used for evaluation for MLLM-I, contains **5,400 samples** (2,491 text-only and 2,909 multimodal inputs). We will correct this in the manuscript to clearly state that MLLM-I were evaluated on the full OpenING benchmark.

---

> > ### Comment · Reviewer_QUx9 · 2025-11-12
> >
> > Thank you for the author's response. After reading, I am satisfied with the author's response. Even though I have seen different opinions from other reviewers, I still have a positive signal for this article. Good luck to you!

---

> > > ### Author Response · Authors · 2025-11-12
> > >
> > > Thank you for your response. We are glad that we addressed your concerns. If you have further questions, please feel free to reach out.

---

### Meta-Review · Area_Chair_EvNj · 2025-12-16

**Summary:**

The reviewers's concerns can be summarized as follows:

1. Latency and computational cost: Reviewers ByY3 and 25UT raised concerns about the latency introduced by tool scheduling and test-time scaling.

2. Benchmark and evaluation: Reviewers 4c1Z and 25UT questioned if the small size of LLMI-Bench and RL datasets is enough for robust conclusion.

3. Novelty: Reviewer 4c1Z viewed the recipe as a combination of known techniques.

**Reviewer Concerns:**

The authors addressed many of the concerns, including latency and benchmark size. Also, the authors provided additional failure analysis, and explanation on reward hacking. These were mostly acknowledged by the reviewers. However, there are remaining concerns about latency and novelty.

**Reviewer Scores:**

This paper receives an initial rating of (2, 4, 4, 8). Based on the rebuttal and discussion, the AC anticipates final scores of (3, 4, 4, 8), which is slightly negative.

---

### Decision · Program_Chairs · 2026-01-26

Reject